# Z-Increments Online Supervisory System Based on Machine Vision for Laser Solid Forming

**DOI:** 10.3390/mi14081558

**Published:** 2023-08-04

**Authors:** Junhua Wang, Junfei Xu, Yan Lu, Tancheng Xie, Jianjun Peng, Junliang Chen

**Affiliations:** 1School of Mechanical and Electrical Engineering, Henan University of Science and Technology, Luoyang 471003, China; mecha_xjf@163.com (J.X.); xietc@haust.edu.cn (T.X.); pjjsdu@163.com (J.P.); 2Henan Intelligent Manufacturing Equipment Engineering Technology Research Center, Luoyang 471003, China; 3Henan Engineering Laboratory of Intelligent Numerical Control Equipment, Luoyang 471003, China; 4School of Materials Science and Engineering, Henan University of Science and Technology, Luoyang 471023, China; luyan@haust.edu.cn; 5College of Food and Bioengineering, Henan University of Science and Technology, Luoyang 471023, China

**Keywords:** machine vision, Z-increments, off-focus amount, closed-loop control system

## Abstract

An improper Z-increment in laser solid forming can result in fluctuations in the off-focus amount during the manufacturing procedure, thereby exerting an influence on the precision and quality of the fabricated component. To solve this problem, this study proposes a closed-loop control system for a Z-increment based on machine vision monitoring. Real-time monitoring of the precise cladding height is accomplished by constructing a paraxial monitoring system, utilizing edge detection technology and an inverse perspective transformation model. This system enables the continuous assessment of the cladding height, which serves as a control signal for the regulation of the Z-increments in real-time. This ensures the maintenance of a constant off-focus amount throughout the manufacturing process. The experimental findings indicate that the proposed approach yields a maximum relative error of 1.664% in determining the cladding layer height, thereby enabling accurate detection of this parameter. Moreover, the real-time adjustment of the Z-increment quantities results in reduced standard deviations of individual cladding layer heights, and the height of the cladding layer increases. This proactive adjustment significantly enhances the stability of the manufacturing process and improves the utilization of powder material. This study can, therefore, provide effective guidance for process control and product optimization in laser solid forming.

## 1. Introduction

Laser solid forming (LSF) is a promising advanced digital additive manufacturing methodology. It seamlessly integrates the advantages of unrestricted solid shaping from rapid prototyping alongside high-performance cladding deposition facilitated by synchronous powder feeding laser cladding [1]. Due to its inherent benefits such as cost-effectiveness, reduced cycle time, exceptional performance, and rapid response capability, LSF has gained substantial traction in various industries, including the aerospace, marine, automotive, and defense sectors, in recent years [2,3]. The effect of LSF forming during the manufacturing process is influenced by a number of factors. Fluctuating parameters and environmental changes during the forming process can cause the cladding height to fluctuate away from the set value. The conventional approach of employing fixed Z-increments during the laser solid forming process has been shown through extensive research to result in substantial variations in the off-focus amount due to fluctuations in the cladding height. These variations, in turn, have been empirically established to exert a direct influence on the dimensional accuracy and mechanical characteristics of the fabricated component [4,5,6,7]. Consequently, the preservation of a consistent off-focus level throughout the forming procedure assumes critical importance in safeguarding the dimensional accuracy and quality of the fabricated part. By implementing real-time monitoring of cladding height variations and establishing a closed-loop Z-increments control system, significant enhancements can be achieved in both the accuracy and quality of the forming process for the part.

A great deal of research is currently being carried out on real-time monitoring and process control of additive manufacturing forming processes. Chen B et al. [8] used a CCD camera to capture images of the melt pool and explored the effect of different process parameters on the melt pool area. It was demonstrated that different types of defects could be accurately detected by analyzing the melt pool area. Binega et al. [9] used a line laser scanner to scan the deposition layer profile in real time to extract the deposition layer geometry. Continuous monitoring of the deposition layer profile of the DED process is achieved by comparing the real-time data with the ideal profile. Takushima et al. [10] proposed an online monitoring system for the deposition layer height of laser-fused wire with a wire feed rate feedback control system. The method achieves high accuracy measurement of the deposition layer height by means of an imaging system and an oblique illumination system for the projected beam. The wire feeding rate is controlled according to the measured deposition layer height to maintain the gap between the wire feeding head and the feeding wire in the optimum zone. Zhang Bi et al. [11] used a coaxial high-speed camera to capture melt pool images and designed a Convolutional neural network model to learn melt pool features with a classification accuracy of 91.2% for porosity detection. Farshdianfar et al. [12] developed an infrared imaging system to monitor the melt pool temperature and cooling rate during the cladding process. Using the surface temperature as the feedback signal, a novel feedback PID controller was developed to control the cooling rate using the correlation between the cooling rate, travel speed, and the microstructure of the clad layer. Fleming et al. [13] monitored the morphology of each layer of the SLM process before and after processing by an inline coherent imaging system, identifying bumps and depressions, remelting the raised areas, and filling the depressed areas, enabling artificial closed-loop control of the surface quality of the solidified layer. Huang et al. [14] used a thermal image to monitor the temperature distribution of the solidification layer of the SLM process, established the relationship between scanning speed and temperature distribution, and maintained a stable solidification layer temperature by adjusting the scanning rate in real time to achieve closed-loop control of the SLM process. Numerous researchers have conducted extensive research to improve the quality of forming products for additive manufacturing [15]. The advancement of monitoring tools and control methodologies has played a pivotal role in the notable progress achieved in the realm of additive manufacturing process control. Nevertheless, a majority of the aforementioned studies primarily concentrated on optimizing the forming quality by controlling process parameters like laser power and scanning speed. In the present investigation of laser solid forming for thin-walled components, a predominant approach involves employing a constant Z-increments and often incorporating a negative off-focus amount to induce a self-healing effect on the morphological attributes. However, the adjustment of the off-focus amount quantity in this manner proves to be inefficient and fails to effectively address the challenge of achieving optimal alignment between the layer height of thin-walled parts and the Z-increments during the cladding process.

To address the challenge of achieving appropriate alignment between the layer height and the Z-increments in laser solid forming fabrication of thin-walled parts, and ensure a constant off-focus amount throughout the manufacturing process, in the present study, we developed an off-axis camera monitoring system that leveraged edge detection techniques and an inverse perspective transformation model to facilitate real-time detection of the cladding height. By utilizing the cladding height as a feedback signal, the detected cladding height served as a control parameter for the robot to dynamically adjust the Z-increments. This control mechanism ensures the maintenance of a consistent off-focus amount throughout the manufacturing process, effectively mitigating potential deviations. The present research study presents a valuable contribution by offering effective guidance in achieving a precise alignment between the layer height and Z-increments, which plays a role in regulating the quality and accuracy of the forming process.

## 2. Design of an Off-Axis Camera Monitoring System

### 2.1. Hardware System Construction

The off-axis camera monitoring system consisted of a Basler industrial camera, filter components, and a workstation. The industrial camera was fixed horizontally to the side axis bracket, thereby ensuring that the optical axis of the camera remained parallel to the substrate. The filter component consisted of a filter lens, a neutral attenuator, and a protective lens. The light emitted from the laser solid forming processing site consisted of multiple wavelengths, intertwining cladding layer information with metal vapor and splashes. Filter lenses facilitated the transmission of specific wavelengths of light while eliminating interference from other wavelength radiations. This capability reduced stray light’s impact on cladding layer images, ensuring a clear imaging of the cladding layer. Simultaneously, the filter lenses could reduce the entry of high-energy laser beams (at 1070 nm) into the CCD camera’s photosensitive sensor, thus mitigating the risk of damaging the sensor. Through literature analysis and comparative testing, this study opted for infrared filter lenses with a range of 800–2500 nanometers. The function of the neutral attenuator laid in its ability to effectively diminish the intensity of light passing through it, thereby mitigating the risk of saturation in the photosensitive element caused by high levels of radiation during the processing stage. For this study, two central attenuation lenses with 10% light transmission were chosen. The laser solid forming process entailed the utilization of a significant quantity of high-temperature metal particles. To safeguard critical components such as cameras and lenses from potential harm or impairment, protective lenses were purposefully engineered, and the adverse consequences stemming from the presence of high-temperature metal particles could be effectively mitigated. In this study, a 2 mm thick quartz glass plate was chosen as the protective lens. The structure and mounting sequence of the filter assembly is shown in Figure 1.

The industrial camera was connected to the workstation and transmitted the captured images in real time to the workstation for image processing. The schematic diagram of the side axis camera monitoring system is shown in Figure 2a, and the site installation diagram is shown in Figure 2b.

### 2.2. Image Coordinate System Transformation

In order to monitor the cladding layer height in real time through the off-axis camera, it was necessary to transform the coordinates of the images collected by the camera to obtain the real height value of the cladding layer.

#### 2.2.1. Perspective Projection Model

The camera projected a three-dimensional scene onto the camera’s two-dimensional plane through an imaging lens. The basic principle of the perspective projection was by converting the coordinates of objects in a three-dimensional scene to coordinates on a two-dimensional plane. The camera perspective projection model consisted of four coordinate systems: the world coordinate system ($O_{W}-X_{W}Y_{W}Z_{W}$), which served as a reference in the environment to describe the position of any object; the camera coordinate system ($O_{C}-X_{C}Y_{C}Z_{C}$), with the camera optical center $O_{C}$ as the origin and the camera optical axis $Z_{C}$ defining its direction; the image coordinate system ($O_{\mathrm{xy}}-xy$), with the intersection of the camera optical axis and the image plane as the origin; and the pixel coordinate system ($O_{uv}-uv$), with the top-left corner of the image as the origin and pixels as the units. The transformation relationships between the coordinate points are illustrated in Figure 3, where Point P represents a point on the cladding layer, and the distance between the camera coordinate system and the image coordinate system origin $O_{C}O_{xy}$ is denoted as f, which represents the camera focal length. $P(X_{C},Y_{C},Z_{C})$ denotes the coordinates of point P in the world coordinate system, p(x,y) represents the projected coordinates of point P in the image coordinate system, and p(u,v) represents the pixel coordinates of point P in the pixel coordinate system.

In the transformation between the world coordinate system and the camera coordinate system, the distances, angles, and parallelism of points remained invariant. The transformation from the world coordinate system to the camera coordinate system comprised translation and rotation transformations, involving solely displacement and rotation without any scaling or non-rigid deformations. Hence, the transformation between the world coordinate system and the camera coordinate system was considered a rigid transformation. Utilizing the rotation matrix R and translation vector T, the transformation of coordinate point P between coordinate systems was expressed as follows:(1)$$\left[\begin{array}{c} X_{C}\\ Y_{C}\\ Z_{C}\\ 1 \end{array}\right]=\left[\begin{array}{cc} R & T\\ 0 & 1 \end{array}\right]\left[\begin{array}{c} X_{W}\\ Y_{W}\\ Z_{W}\\ 1 \end{array}\right]$$
where the rotation matrix *R* is an 3×3 orthogonal unit matrix, dimensionless; *T* is a three-dimensional translation matrix, which has units in mm.

The transformation of the camera coordinate system to the image coordinate system was a perspective projection, converting the coordinate point P from three-dimensional to two-dimensional. From the proportional relationship, it could be obtained:(2)$$Z_{C}\left[\begin{array}{c} x\\ y\\ 1 \end{array}\right]=\left[\begin{array}{cccc} f & 0 & 0 & 0\\ 0 & f & 0 & 0\\ 0 & 0 & 1 & 0 \end{array}\right]\left[\begin{array}{c} X_{C}\\ Y_{C}\\ Z_{C}\\ 1 \end{array}\right]$$
where f indicates camera focal length; x and y represent the horizontal coordinate and vertical coordinates of the projection of coordinate point P onto the image coordinate system; the units of f, x, and y are mm.

As can be seen from Figure 3b, the image coordinate system and pixel coordinate system are both in the imaging plane but have different origins and units of measure. The conversion relation of coordinate points is:(3)$$\left[\begin{array}{c} u\\ v\\ 1 \end{array}\right]=\left[\begin{array}{ccc} \frac{1}{dx} & 0 & u_{0}\\ 0 & \frac{1}{dy} & v_{0}\\ 0 & 0 & 1 \end{array}\right]\left[\begin{array}{c} x\\ y\\ 1 \end{array}\right]$$
where *u* and *v* denote the horizontal coordinate and vertical coordinates in the pixel coordinate system, respectively; the unit is px. *dx* and *dy* represent the physical size of each pixel in mm/px; $u_{0}$ and $v_{0}$ are the origin positions; the unit is px.

From Equations (1)–(3), the transformation relationship between the world coordinate system and the pixel coordinate system can be modelled as:(4)$$Z_{C}\left[\begin{array}{c} u\\ v\\ 1 \end{array}\right]=\left[\begin{array}{cccc} f_{x} & 0 & u_{0} & 0\\ 0 & f_{y} & v_{0} & 0\\ 0 & 0 & 1 & 0 \end{array}\right]\left[\begin{array}{cc} R & T\\ 0 & 1 \end{array}\right]\left[\begin{array}{c} X_{W}\\ Y_{W}\\ Z_{W}\\ 1 \end{array}\right]$$
where $f_{x}=f/dx$ and $f_{y}=f/dy$ are the scale factors for the u axis and v axis, respectively. $f_{x},f_{y},u_{0},v_{0}$ are internal camera parameters; *R* and *T* are external camera parameters. Simplifying the model representation:(5)$$M_{1}=\left[\begin{array}{cccc} f_{x} & 0 & u_{0} & 0\\ 0 & f_{y} & v_{0} & 0\\ 0 & 0 & 1 & 0 \end{array}\right],{\;M}_{2}=\left[\begin{array}{cc} R & T\\ 0 & 1 \end{array}\right]$$

$M_{1}$ is the camera internal parameter matrix, and $M_{2}$ is the camera external parameter matrix.

#### 2.2.2. Camera Calibration

To effectively establish the mapping relationship between two-dimensional and three-dimensional images, it was imperative to incorporate the projection characteristics inherent in the transformation process from the camera to the image. This entailed solving for the pertinent parameters of this model by utilizing the corresponding relationship between the mathematical model of camera imaging and the underlying coordinate system. This procedure is commonly referred to as camera calibration [16]. In the present study, the calibration process of the CCD camera involved the utilization of a circular point calibration plate. The calibration plate consisted of 7×7 circular points, each possessing a diameter of 3.5 mm. These circular points were positioned at a uniform center distance of 7 mm from one another. Additionally, the circular point calibration plate featured a square inner frame measuring 56 mm in dimension.

Formula (4) represents the transformation relationship between the world coordinate system and the pixel coordinate system. $M_{1}$ denotes the intrinsic matrix, which is solely dependent on the camera’s intrinsic properties and internal structure. $M_{2}$ is determined by the mapping relationship between the world coordinate system and the camera coordinate system. The camera calibration process involves the estimation of $M_{1}$ and $M_{2}$.

Due to lens imperfections, it was impossible for the camera’s imaging model to achieve an ideal state, leading to distortions in the captured images. Nonlinear distortions mainly consisted of radial distortion and tangential distortion. To enhance the precision of camera calibration, this study not only obtained the radial distortion coefficient $k_{1},k_{2},k_{3}$ but also derived two tangential distortion coefficients $p_{1},p_{2}$ during the calibration process. The nonlinear distortion model is represented as follows:(6)$$\left\{\begin{array}{l} x_{u}=x_{d}+\delta_{x}(x_{d},y_{d})\\ y_{u}=y_{d}+\delta_{y}(x_{d},y_{d}) \end{array}\right.$$
where $\left(x_{u},y_{u}\right)$ represents the ideal coordinate values of the image point, $\left(x_{d},y_{d}\right)$ denotes the actual coordinate values of the image point, and $\delta_{x},\delta_{y}$ represents the nonlinear distortion values. The nonlinear distortion expression employed in this study is as follows:(7)$$\left\{\begin{array}{l} \delta_{x}(x_{d},y_{d})=x_{d}(k_{1}r_{d}^{2}+k_{2}r_{d}^{2}+k_{3}r_{d}^{2})+[p_{1}(3x_{d}^{2}+y_{d}^{2})+2p_{2}x_{d}y_{d}]\\ \delta_{y}(x_{d},y_{d})=y_{d}(k_{1}r_{d}^{2}+k_{2}r_{d}^{2}+k_{3}r_{d}^{2})+[p_{2}(3x_{d}^{2}+y_{d}^{2})+2p_{1}x_{d}y_{d}] \end{array}\right.$$
where $r_{d}^{2}=x_{d}^{2}+y_{d}^{2}$.

After calibration and calculation, the camera internal parameter matrix $M_{1}$ is obtained as:$$M_{1}=\left[\begin{array}{cccc} 5598.16 & 0 & 530.38 & 0\\ 0 & 5597.33 & 404.37 & 0\\ 0 & 0 & 1 & 0 \end{array}\right]$$

The rotation matrix *R* and translation vector *T* in the camera external parameter matrix are:$$R=\left[\begin{array}{ccc} 1 & -0.008 & -0.016\\ 0.007 & 0.98 & -0.029\\ 0.017 & 0.029 & 0.98 \end{array}\right],T=\left[\begin{array}{c} -1.52\\ -6.055\\ 392.78 \end{array}\right]$$

#### 2.2.3. Inverse Perspective Transformation

The establishment of transformation relationships among different coordinate systems was accomplished through camera calibration, which involved determining the internal and external matrix parameters of the camera. Visual measurement entailed an inverse perspective transformation process, distinct from the perspective transformation process described earlier. In this context, visual measurement involved the conversion of the image pixel dimensions of the target object from the image coordinate system to the world coordinate system, so as to obtain the actual size of the measured object. The inverse perspective transformation entailed the utilization of known camera internal parameter matrix $M_{1}$, camera external parameter matrix $M_{2}$, and image pixel coordinate points; Equation (4) is transformed into a linear equation about the three unknowns of $X_{W},Y_{W},Z_{W}$; and there is no unique solution to the system of equations. To ensure the existence of a unique solution for the aforementioned equation, the imposition of a constraint became necessary. This constraint facilitated the achievement of the inverse perspective transformation, enabling the conversion of two dimensions image pixel coordinate points to three-dimensional world coordinate points.

In this study, the relative positional relationship between the camera and the cladding layer did not change as the experiment proceeded. During the camera calibration process, the calibration plate and the cladding layer were on the same plane, and the plane $X_{W}Y_{W}$ in the world coordinate system coincided with the plane of the cladding layer, i.e., $Z_{W}=0$. Therefore, by adding this constraint condition, the inverse perspective transformation of the camera was created.

Combining the camera perspective transformation model, let the rotation matrix *R* and translation matrix *T* be:(8)$$R=\left[\begin{array}{ccc} r_{11} & r_{12} & r_{13}\\ r_{21} & r_{22} & r_{23}\\ r_{31} & r_{32} & r_{33} \end{array}\right],\;T=\left[\begin{array}{c} t_{1}\\ t_{2}\\ t_{3} \end{array}\right]$$

Then, Equation (4) is converted to:(9)$$Z_{C}\left[\begin{array}{c} u\\ v\\ 1 \end{array}\right]=\left[\begin{array}{cccc} f_{x} & 0 & u_{0} & 0\\ 0 & f_{y} & v_{0} & 0\\ 0 & 0 & 1 & 0 \end{array}\right]\left[\begin{array}{cccc} r_{11} & r_{12} & r_{13} & t_{1}\\ r_{21} & r_{22} & r_{23} & t_{2}\\ r_{31} & r_{32} & r_{33} & t_{3}\\ 0 & 0 & 0 & 1 \end{array}\right]\left[\begin{array}{c} X_{W}\\ Y_{W}\\ Z_{W}\\ 1 \end{array}\right]$$

As the plane of the world coordinate system $X_{W}Y_{W}$ coincides with the plane in which the fused layer is located during camera calibration, such that $Z_{W}=0$, Equation (7) is simplified by matrix transformation as:(10)$$Z_{C}\left[\begin{array}{c} u\\ v\\ 1 \end{array}\right]=\left[\begin{array}{ccc} f_{x} & 0 & u_{0}\\ 0 & f_{y} & v_{0}\\ 0 & 0 & 1 \end{array}\right]\left[\begin{array}{ccc} r_{11} & r_{12} & t_{1}\\ r_{21} & r_{22} & t_{2}\\ r_{31} & r_{32} & t_{3} \end{array}\right]\left[\begin{array}{c} X_{W}\\ Y_{W}\\ 1 \end{array}\right]$$

Then, it follows that:(11)$$Z_{C}\left[\begin{array}{c} u\\ v\\ 1 \end{array}\right]=\left[\begin{array}{ccc} f_{x}r_{11}+u_{0}r_{31} & f_{x}r_{12}+u_{0}r_{32} & f_{x}t_{1}+u_{0}t_{3}\\ f_{y}r_{21}+v_{0}r_{31} & f_{y}r_{22}+v_{0}r_{32} & f_{y}t_{2}+v_{0}t_{3}\\ r_{31} & r_{32} & t_{3} \end{array}\right]\left[\begin{array}{c} X_{W}\\ Y_{W}\\ 1 \end{array}\right]$$

Simplification of Equation (9)
(12)$$H=\left[\begin{array}{ccc} f_{x}r_{11}+u_{0}r_{31} & f_{x}r_{12}+u_{0}r_{32} & f_{x}t_{1}+u_{0}t_{3}\\ f_{y}r_{21}+v_{0}r_{31} & f_{y}r_{22}+v_{0}r_{32} & f_{y}t_{2}+v_{0}t_{3}\\ r_{31} & r_{32} & t_{3} \end{array}\right]$$

Substituting Equation (10) into (9), the inverse perspective transformation model is obtained as follows:(13)$$\left[\begin{array}{c} X_{W}\\ Y_{W}\\ 1 \end{array}\right]=Z_{C}H^{-1}\left[\begin{array}{c} u\\ v\\ 1 \end{array}\right]$$

Utilizing the established inverse perspective transformation model, the coordinates of a point within the pixel coordinate system are utilized to derive its corresponding coordinates in the world coordinate system. This process relies on the camera’s calibrated internal and external parameters, ultimately yielding the accurate size of the cladding layer within the world coordinate system.

### 2.3. Image Region of Interest Extraction

The initial image of the cladding layer possessed dimensions of 1280 × 1024 pixels, encompassing various elements such as the table, substrate, thin-walled components, and residual unmelted powder. Given the presence of numerous pixel points in the original image that were irrelevant to the study and potentially impeded the extraction of valuable information, it became necessary to isolate the region of interest (ROI) within the original image. In this study, the ROI was extracted from the original image, resulting in a rectangular area defined as [350:650, 70:1020]. The dimensions of the extracted image measured 950×300 pixels, as depicted in Figure 4. The extracted image showed mainly the cladding and the substrate, eliminating the interference of redundant pixel points and facilitating further processing of the image.

### 2.4. Cladding Layer Contour Extraction

In this study, we addressed the issue of edge blurring encountered in the conventional Canny algorithm, which employed Gaussian filtering for noise reduction. To preserve the integrity of edges, we opted for bilateral filtering as an alternative to Gaussian filtering. Furthermore, we tackled the problem of artificially defined thresholds by employing an enhanced Otsu algorithm to derive image segmentation thresholds.

#### 2.4.1. Image Filtering

The bilateral filter incorporated both spatial domain information of pixel points and value domain information based on a Gaussian filter framework. Traditional filtering methods tended to introduce edge blurring during gradual image transformations [17]. In contrast, bilateral filtering considered both the Euclidean distance between pixels and the gray value information of the image, enabling preservation of edge details while accomplishing denoising. Specifically, when the pixel values on either side of an edge differed, weights were diminished to give greater influence to neighboring pixels on the similar side, effectively preventing edge blurring. The bilateral filter pattern is shown below:(14)$$\overset{\wedge }{f}(i,j)=\frac{\sum_{(m,n)\;\in \;\Omega_{r,i,j}}\omega_{d}(m,n)\omega_{r}(m,n)f(m,n)}{\sum_{(m,n)\;\in \;\Omega_{i,j}}\omega_{d}(m,n)\omega_{r}(m,n)}$$
(15)$$\omega_{d}(m,n)=\exp(-\frac{{\left(i-m\right)}^{2}+{\left(j-n\right)}^{2}}{2\sigma_{d}^{2}})$$
(16)$$\omega_{r}(m,n)=\exp(-\frac{f(i,j)+f{\left(m,n\right)}^{2}}{2\sigma_{r}^{2}})$$
where f(m,n) is the gray value of the input image at coordinate (*m*,*n*); $\overset{\wedge }{f}(i,j)$ is the gray value of the filtered image at coordinate (*i*,*j*); *r* is the filter window radius; $\Omega_{\;r,i,j}$ is the set of coordinates of pixels in a square region with (*i*,*j*) as the center and sides of (2*r* + 1); $\omega_{d}(m,n)$ and $\omega_{r}(m,n)$ are the spatial weights and gray similarity weights at coordinates (*m*,*n*), respectively; $\sigma_{d}$ and $\sigma_{r}$ are spatial standard deviation and gray standard deviation, respectively.

When bilaterally filtering, as the strong noise differed significantly from the grey value of the central pixel, the obtained grey similarity was weighted more heavily, so this strong noise was retained as edge [18]. To address this issue, this study used a combination of adaptive median filtering, which had a high ability to remove strong noise and retain image information. The adaptive median filtering technique enabled dynamic adjustment of the filter window size based on varying noise densities encountered.

The steps of adaptive median filtering were as follows:If $f_{\min}<f_{med}<f_{\max}$, go to step 2, otherwise increase window size $S_{xy}$. If $S_{xy}\leq S_{\max}$, repeat step 1, otherwise output $f_{med}$.If $f_{\min}<f_{ij}<f_{\max}$, output $f_{ij}$, otherwise output $f_{med}$.

Where $S_{xy}$ is a window centered on the coordinates (*x*,*y*); $S_{\max}$ is the maximum size allowed for the window; $f_{ij}$ is the gray value of the coordinate (*x*,*y*); $f_{\min}$, $f_{med}$, and $f_{\max}$ are the minimum grey value, the median grey value, and the maximum grey value of $S_{xy}$, respectively.

#### 2.4.2. Gradient Amplitude Calculation

Calculation of gradient amplitude after image filtering. The 45° and 135° directional gradient amplitudes are introduced in the 3×3 neighborhood, and the four directional gradient amplitudes are first obtained by means of the Sobel gradient template. The direction gradient templates for each direction are as follows:

Vertical (x) direction template:$$\left[\begin{array}{ccc} -1 & -2 & -1\\ 0 & 0 & 0\\ 1 & 2 & 1 \end{array}\right]$$

Horizontal (y) direction template
$$\left[\begin{array}{ccc} -1 & 0 & 1\\ -2 & 0 & 2\\ -1 & 0 & 1 \end{array}\right]$$

45° orientation template
$$\left[\begin{array}{ccc} -2 & -1 & 0\\ -1 & 0 & 1\\ 0 & 1 & 2 \end{array}\right]$$

135° orientation template
$$\left[\begin{array}{ccc} 0 & 1 & 2\\ -1 & 0 & 1\\ -2 & -1 & 0 \end{array}\right]$$

The formula for calculating the gradient amplitude is as follows:(17)$$M(x,y)=\sqrt{G_{xy}^{2}(x,y)+G_{bevel}^{2}(x,y)}$$
(18)$$G_{xy}^{2}=G_{x}^{2}(x,y)+G_{y}^{2}(x,y)$$
(19)$$G_{bevel}^{2}=G_{45{}^{\circ}}^{2}(x,y)+G_{135{}^{\circ}}^{2}(x,y)$$
where $G_{x}$ represents the gradient magnitude in the x-direction, $G_{y}$ represents the gradient magnitude in the y-direction, $G_{45{}^{\circ}}^{}$ represents the gradient magnitude at a 45° angle, and $G_{135{}^{\circ}}^{}$ represents the gradient magnitude at a 135° angle.

#### 2.4.3. Improved Otsu Algorithm for Threshold Segmentation

The Otsu algorithm constitutes a technique employed to determine the optimal threshold value by performing calculations on the image histogram as a fundamental basis. By leveraging the statistical characteristics of the histogram, this algorithm aims to identify the threshold value that maximizes the inter-class variance, thereby effectively segmenting the image into distinct regions. The Otsu algorithm initiates by determining an optimal threshold value, denoted as k, to facilitate the segmentation of the image into foreground and background regions. Subsequently, the algorithm computes the inter-class variance between these segmented regions. Notably, a higher disparity between the foreground and background intensities results in an augmented inter-class variance, indicative of an improved thresholding outcome. The classes square error $\sigma_{B}^{2}$ is defined as shown in Equation (18):(20)$$\sigma_{B}^{2}=P_{1}{\left(m_{1}-m_{G}\right)}^{2}+P_{2}{\left(m_{2}-m_{G}\right)}^{2}$$
where $P_{1},P_{2}$ are the probability that a pixel will be assigned to the foreground and background regions, respectively; $m_{1},m_{2},m_{G}$ are the foreground, background, and global average greyscale values, respectively.

To evaluate the quality of the processed image at threshold *k*.
(21)$$\eta =\frac{\sigma_{B}^{2}}{\sigma_{G}^{2}}$$
where $\sigma_{G}^{2}$ is the global variance. From Equation (19), as $\sigma_{G}^{2}$ is a constant, and $\sigma_{B}^{2}$ is a divisibility measure between classes, η is also a divisibility measure, and the two are maximally equivalent.

The Otsu algorithm demonstrated optimal performance when employed for segmenting images exhibiting prominent bimodal peaks within the histogram. However, its efficacy diminished when applied to labeled images containing a sparse distribution of target pixel points. In such cases, the segmentation threshold tended to exhibit a strong bias towards background regions characterized by a substantial proportion of pixels and significant intra-class variance [19]. The Otsu algorithm primarily determined thresholds by maximizing the interclass variance. Leveraging this characteristic, an enhanced segmentation outcome could be achieved by incorporating additional considerations, such as the grey value height within the threshold neighborhood and the average grey difference within the region. By incorporating these factors, the algorithm accentuated the discernibility of low grey value troughs, leading to improved differentiation between the foreground and background regions. The improved formula for the classes square error is:(22)$$\sigma_{B}^{2}=(1-\sum_{i=k-n}^{k+n}P_{i}{)}^{a}(P_{1}{\left(m_{1}-m_{G}\right)}^{2}+P_{2}{\left(m_{2}-m_{G}\right)}^{2}+\left|m_{1}-m_{2}\right|)$$
where $\sum_{i=k-n}^{k+n}p_{i}$ is the probability of distribution of all pixels in the gray value interval [*k* − *n*,*k* + *n*]; a is the setting parameter; and a tends to take a larger value when the trough is not evident, making the threshold region trough.

The flowchart for image edge extraction is illustrated in Figure 5.

To extract the pertinent information regarding the cladding layer height, the thresholded image underwent edge detection, specifically targeting the top edge profile of the cladding layer. This extracted profile encapsulated the essential details required for accurately calculating the cladding layer’s height. By employing the edge detection technique, the prominent edges of the cladding layer were detected and extracted, thereby facilitating precise determination of its height through subsequent analysis. The image after the canny operator edge detection process is shown in Figure 6.

### 2.5. Height Calculation and Analysis of Results

#### 2.5.1. Height Calculation

The image after Canny operator edge detection was pixel traversed, the grey value of each pixel was traversed in turn, and the horizontal and vertical coordinates of the pixel with a grey value of 255 were recorded to obtain the horizontal and vertical coordinates of each pixel in the extracted edge profile of the thin-walled part. The vertical coordinate of the topmost contour of the substrate was subtracted from the vertical coordinate of each pixel point at the edge of the contour to obtain the pixel height of the thin-walled part. In this way, the overall pixel heights of the thin-walled parts in layers 5, 10, 15, 20, 25, and 30 were obtained, and the results are shown in Figure 7.

According to the camera calibration results and the inverse perspective transformation model, the coordinate points were transformed with inverse perspective to obtain the actual height of the cladding layer. The heights of the 5th, 10th, 15th, 20th, 25th, and 30th cladding layers are shown in Figure 8.

#### 2.5.2. Height Error Analysis

The cladding height calculation error was obtained by comparing the cladding height calculation results with the height measurement values. Five positions were selected on the uppermost contour of the finished 30-ply thin-walled part, at approximately 4 mm, 20 mm, 35 mm, 48 mm, and 59 mm from the leftmost end in the horizontal direction, and the five selected measurement positions are shown in Figure 9.

Measurements were made using a height micrometer, with the measurement positions sorted from left to right, and the image calculation of the height against the actual measured data is shown in Figure 10.

As illustrated in Figure 9, the analysis reveals a maximum relative error of 1.644% and a minimum relative error of 0.567% between the computed image-based cladding height values and the corresponding actual measured values. This demonstrates the high accuracy achieved in accurately quantifying the cladding height through the proposed methodology. The minimal relative error suggests precise measurement capabilities, thus enabling reliable assessment of the cladding height based on the image analysis. The observed errors can be attributed to several underlying factors. Firstly, inherent characteristics of the CCD camera chip itself may contribute to the issue. When capturing certain objects, inadequate contrast of the edge contour can arise, thus adversely affecting the efficacy of image processing. Secondly, errors within the imaging system can significantly impact the detection accuracy. The resolution of the industrial camera, in particular, plays a crucial role in achieving precise measurements. Additionally, geometric distortion constitutes another influential factor that impairs detection accuracy. Lastly, the presence of vibrations emerges as a prominent contributor to variations in the visual inspection results. Even slight vibrations can result in blurred and distorted images, thereby exerting a detrimental influence on the accuracy of detection.

## 3. Design of Closed-Loop Control System for Z-Increments

When thin-walled parts are formed under constant Z-increment conditions, the off-focus amount will accumulate layer by layer and gradually increase, resulting in a gradual reduction in the height of the thin-walled part, which ultimately leads to a lower total height of the thin-walled part at the corresponding position, an uneven surface of the thin-walled part, and poor-forming dimensional accuracy. To facilitate the attainment of precise and high-quality manufacturing of thin-walled components through LSF technology, this study introduced a Z-increment regulation method. This method is based on monitoring the forming process of thin-walled parts from the side-axis, enabling the determination of the total height of the fabricated thin-walled parts. By leveraging this information, the proposed method regulates the Z-increments, ensuring accurate control of the additive manufacturing process.

In this study, the Z-increments were used to monitor the layer height of thin-walled parts to ensure that the off-focus amount was kept within a constant range during the manufacturing process and to reduce the impact of the off-focus amount on the LSF forming quality, thereby achieving the goal of regulating the shape size and forming quality of thin-walled parts.

To accurately determine the incremental increase in height for each layer of cladding during the forming process of a thin-walled part, a calculation approach was employed. Specifically, the height value at the conclusion of the current layer of cladding was subtracted from the height value prior to the current layer of cladding. By performing this subtraction, the actual layer height for each layer of cladding within the thin-walled part could be ascertained. The calculation formula is shown below:(23)$$h_{n}=H_{n}-H_{n-1}$$
where $h_{n}$ is the height of the nth cladding layer, and $H_{n}$ is the total height of the thin-walled part at the nth layer.

The control system flow is shown in Figure 11.
The image of the cladding layer was captured in real time using a side axis camera and transmitted to the host computer workstation;Image processing of the acquired images, including ROI extraction, image filtering, noise reduction, image thresholding, and edge detection;The camera was calibrated, and the results of the camera calibration were used to perform an inverse perspective transformation of the image pixel points to calculate the cladding height;The layer height of the cladding layer was calculated by Equation (21) and transmitted to the PLC from the host computer as the Z-increments;The PLC sent a command to KUKA Robotics with the Z-increments acquired in step 4, which, in turn, ensured that the off-focus amount remained constant during the manufacturing process;Detected whether the number of layers processed had reached the preset value. If the preset value was reached, the process ended; if the preset value was not reached, the process re-entered step 1.

This study used the monitored clad height as a Z-increment to effectively ensure a constant off-focus amount during the manufacturing process, which, in turn, provided guidance for clad quality regulation.

## 4. Experiments and Analysis of Results

### 4.1. Materials and Setup

All experiments in this study were carried out on a laser solid forming equipment, which consisted of a KUKA robot, a co-flying water cooler, a carrier air powder feeder, and a 3 Kw power All-Light laser and laser head. The laser head was connected to a water cooler to avoid damage to the equipment due to high temperatures during the manufacturing process. The powder feed gas and protective gas were both 99.99% argon with a gas flow rate of 12 L/min. The laser solid forming system is shown in Figure 12.

In the present experiment, a substrate composed of 45 steel, with dimensions measuring 20 mm by 10 mm by 8 mm, was employed. To mitigate any potential temperature-related interferences resulting from multiple cladding, only a single cladding experiment was conducted for each individual substrate. In this study, 17-4PH powder was used as the cladding material, and the chemical composition is shown in Table 1.

### 4.2. Design of Experiments

Comparative experiments using the laser solid forming system were carried out with constant Z-increment cladding and real-time, regulated Z-increment cladding. The constant Z-increments were set at 0.25 mm for the cladding experiments, and the other experimental conditions are shown in Table 2.

### 4.3. Analysis of Experimental Results

The evaluation of the modulation effect in this study focused on assessing the smoothness of each cladding layer’s height. To achieve a more objective and accurate quantification of height smoothness, the standard deviation of the height values associated with each layer was employed as an evaluation metric. The standard deviation formula is shown below:(24)$$\sigma =\sqrt{\frac{1}{N}\sum_{i=1}^{N}{\left(x_{i}-\mu \right)}^{2}}$$
where *N* is the number of data points; *i* is the *i*-th data; and μ is the overall mean.

Each layer of cladding in the experiment was monitored paraxially, the standard deviation of its height value per layer was calculated, and the results of the experiment are shown in Figure 13. The blue line in Figure 13 represents the standard deviation of the height per level for the constant z-increments experiment, and the red line represents the standard deviation of the height per level for the real-time, regulated z-increments experiment.

As can be seen from Figure 13, the standard deviation of the height per level for the real-time regulated Z-increments experiment is significantly smaller than the standard deviation of the height for a constant Z-increments experiment, and this difference becomes larger as the number of levels increases. The analysis suggests that the change in off-focus amount produces cumulative errors as the number of layers increases under constant Z-increments conditions, making the instability of the cladding process worse. Real-time regulation of the Z-increments ensures that the off-focus amount remains constant during the manufacturing process, avoiding the accumulation of errors.

Due to the heat accumulation caused by the continuous cladding process, fluctuations in height standard deviation could occur. The height standard deviation stabilized after the experiment reached 20 layers. The analysis suggested that this phenomenon was due to the “dynamic equilibrium” between the melt pool, the cladding layer, and the substrate at this point, where the heat input and heat transfer became balanced, and the whole process became relatively stable, with height fluctuations stabilizing.

The values of the height of each cladding layer monitored during the experiment are shown in Figure 14.

As can be seen from Figure 14, the height per layer for the real-time, regulated Z-increments experiment is significantly greater than the height of the cladding layer for the constant Z-increments experiment. The analysis concluded that the effect of the off-focus amount on the cladding height was reduced by regulating the Z-increments, increasing the amount of powder entering the melt pool and improving powder utilization. At the same time, the standard deviation of the layer height was calculated. The layer height of the cladding layer in the controlled Z-increments experiment fluctuated less, and the standard deviation of the layer height was 0.015 mm; the layer height of the constant Z-increments experiment fluctuated more, and the standard deviation of the layer height was 0.027 mm. By controlling the z-increments, the layer height of the thin-walled part was relatively stable.

## 5. Conclusions

Inadequate z-increments during laser solid forming could lead to variations in the off-focus amount and thus affect the quality of the formed part. To solve this problem, this study proposed an LSF regulation method based on machine vision technology using a side axis camera to monitor the cladding height in real time and use the real time cladding height as a control signal to regulate the Z-increments. Through experimental verification, the following conclusions were obtained:In this study, an off-axis camera was used to capture the cladding height image in real time, and, after ROI region extraction, edge detection, camera calibration, and inverse perspective transformation, the actual cladding height was obtained. Through experimental verification, the maximum measurement error was 1.664%. This method could measure the cladding layer height more accurately and in real time;This study was based on machine vision using an off-axis camera to measure the cladding height in real time and use the cladding height as a control signal to regulate the z-increments. The results of the comparative experiments showed that the height of the cladding layer was more stable, the forming accuracy was improved, and the powder utilization rate was increased by real-time adjustment of the Z-increments. The results proved that the study could effectively improve the stability of the forming process and provide effective guidance for practical production.

## Figures and Tables

**Figure 1 micromachines-14-01558-f001:**
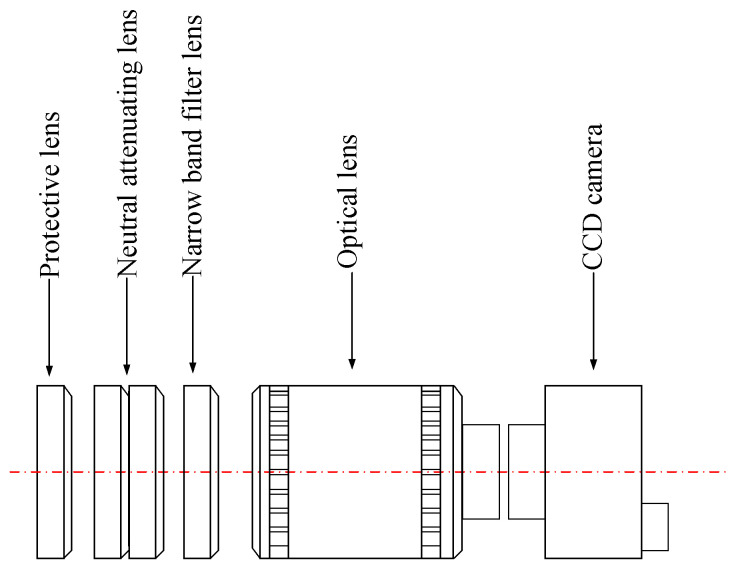
Diagram of the filter components.

**Figure 2 micromachines-14-01558-f002:**
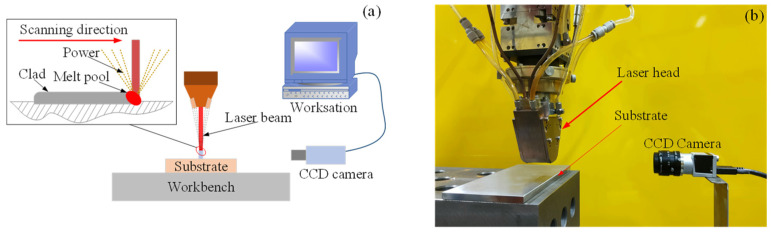
Side axis camera monitoring system. (**a**) Diagram of the off-axis camera monitoring system; (**b**) site installation drawings.

**Figure 3 micromachines-14-01558-f003:**
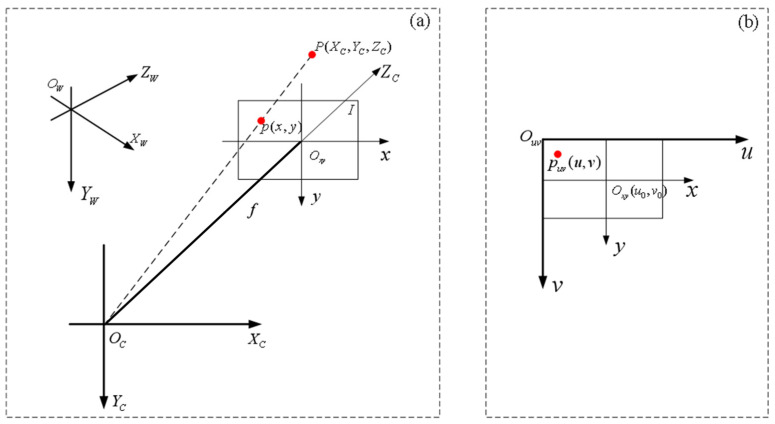
Diagram of the camera perspective projection model. (**a**) World coordinate system and camera coordinate system; (**b**) image coordinate system and pixel coordinate system.

**Figure 4 micromachines-14-01558-f004:**
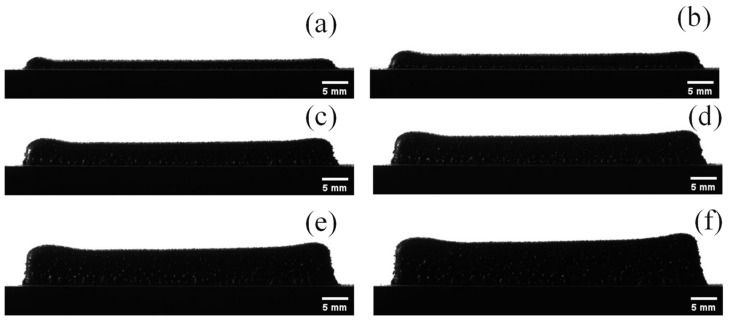
Images after ROI extraction. (**a**) Image after ROI extraction of 5 layers; (**b**) image after ROI extraction of 10 layers; (**c**) image after ROI extraction of 15 layers; (**d**) image after ROI extraction of 20 layers; (**e**) image after ROI extraction of 25 layers; and (**f**) image after ROI extraction of 30 layers.

**Figure 5 micromachines-14-01558-f005:**
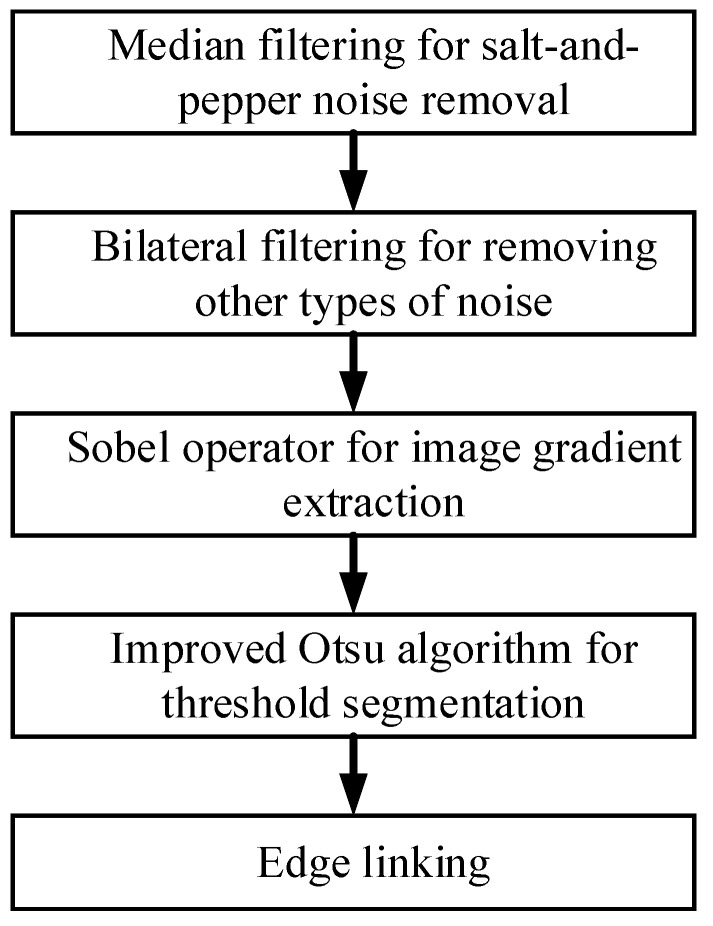
Flowchart of edge extraction.

**Figure 6 micromachines-14-01558-f006:**
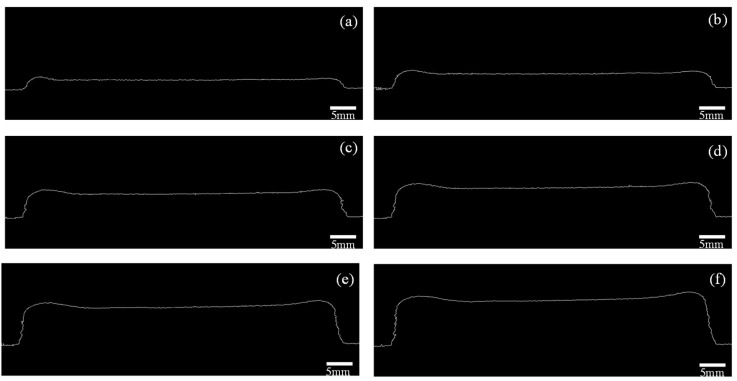
Images after Canny edge detection. (**a**) Image of edge detection of 5 layers; (**b**) image of edge detection of 10 layers; (**c**) image of edge detection of 15 layers; (**d**) image of edge detection of 20 layers; (**e**) image of edge detection of 25 layers; (**f**) image of edge detection of 30 layers.

**Figure 7 micromachines-14-01558-f007:**
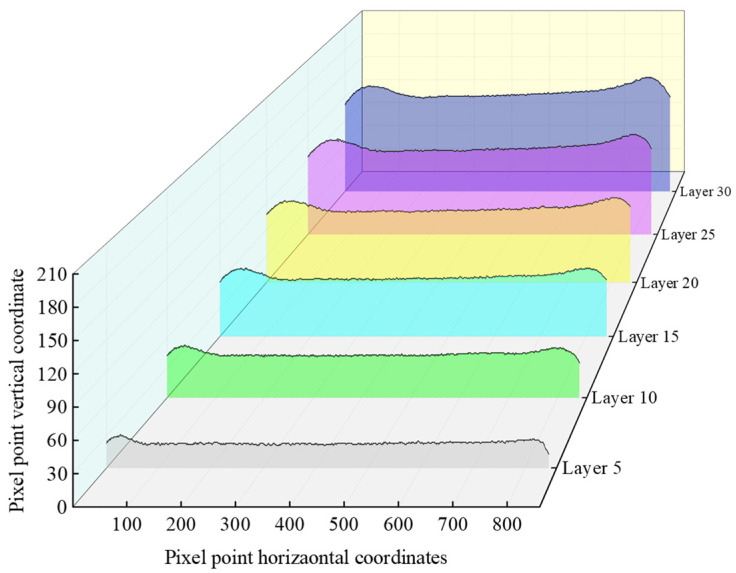
Pixel heights of thin-walled parts.

**Figure 8 micromachines-14-01558-f008:**
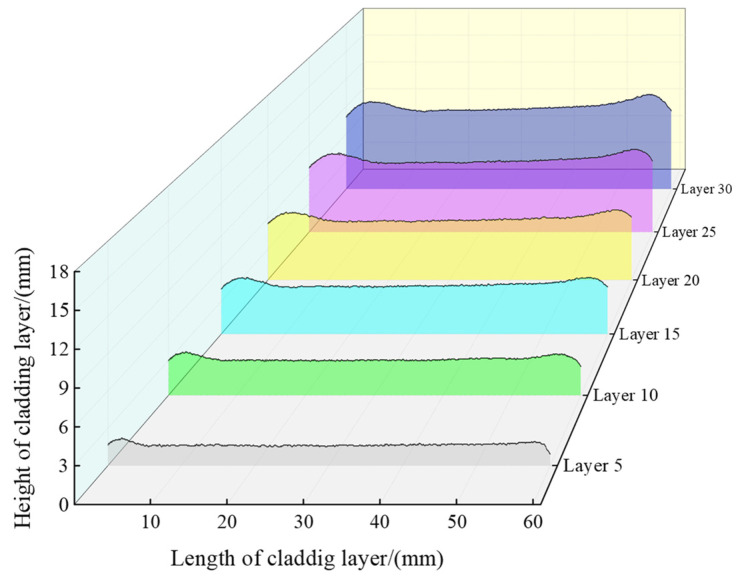
Calculated heights of thin-walled parts.

**Figure 9 micromachines-14-01558-f009:**
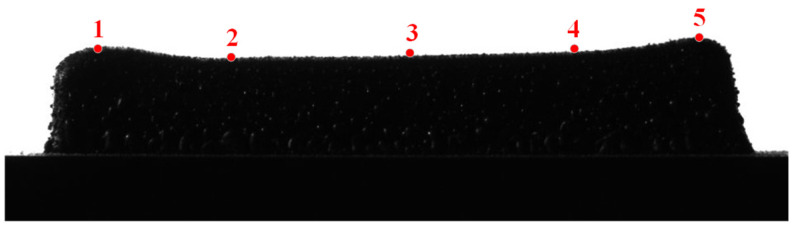
Measurement sites of thin-walled parts. Point 1 is located 4 mm from the leftmost end; Point 2 is located 20 mm from the leftmost end; Point 3 is located 35 mm from the leftmost end; Point 4 is located 48 mm from the leftmost end; Point 5 is located 59 mm from the leftmost end.

**Figure 10 micromachines-14-01558-f010:**
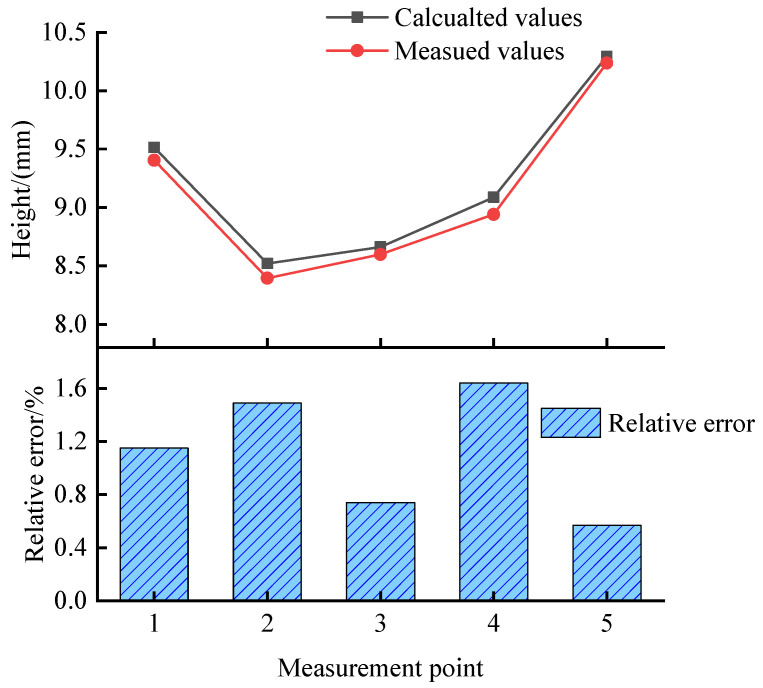
Comparison between calculated and measured values of height.

**Figure 11 micromachines-14-01558-f011:**
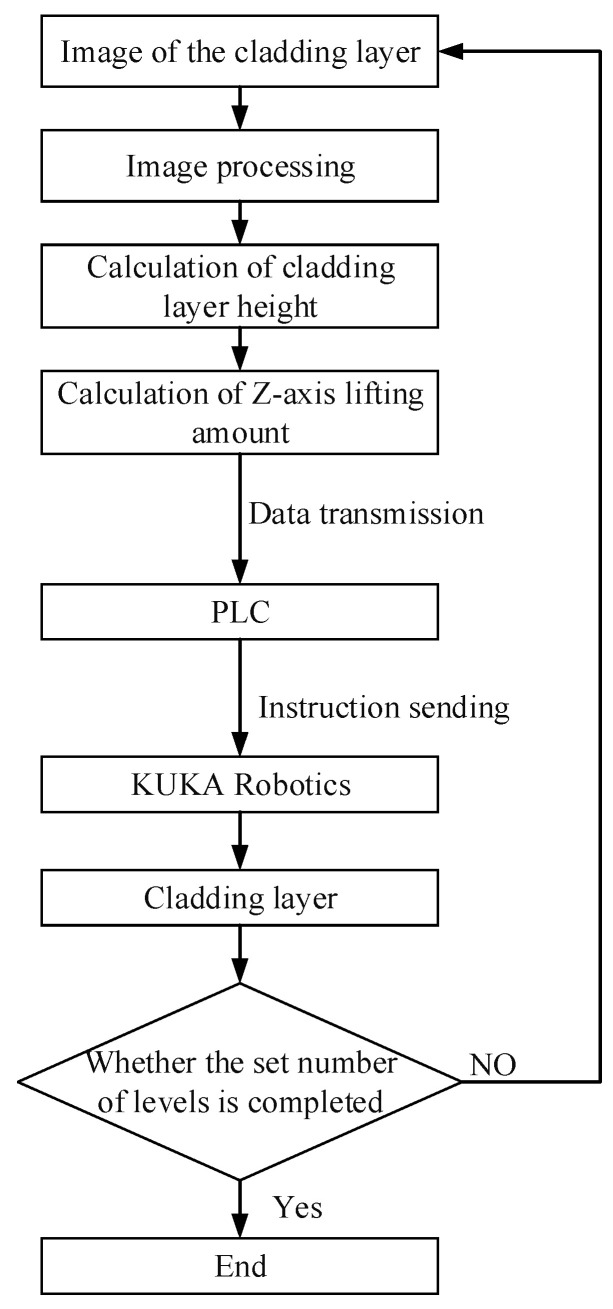
Control flow system diagram.

**Figure 12 micromachines-14-01558-f012:**
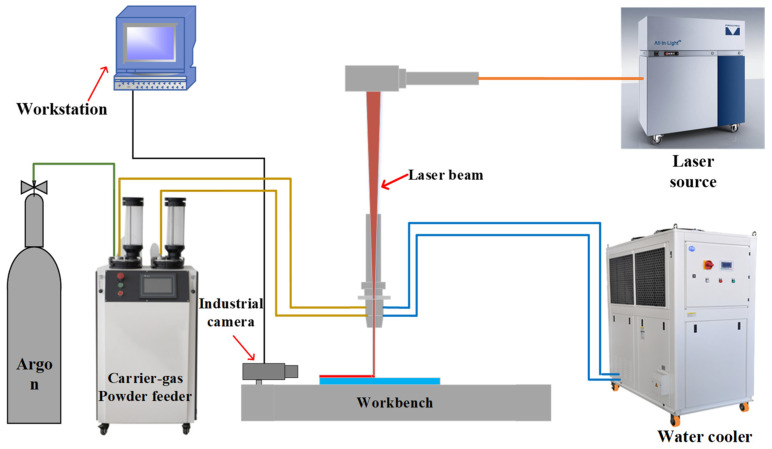
Diagram of the laser stereo forming system.

**Figure 13 micromachines-14-01558-f013:**
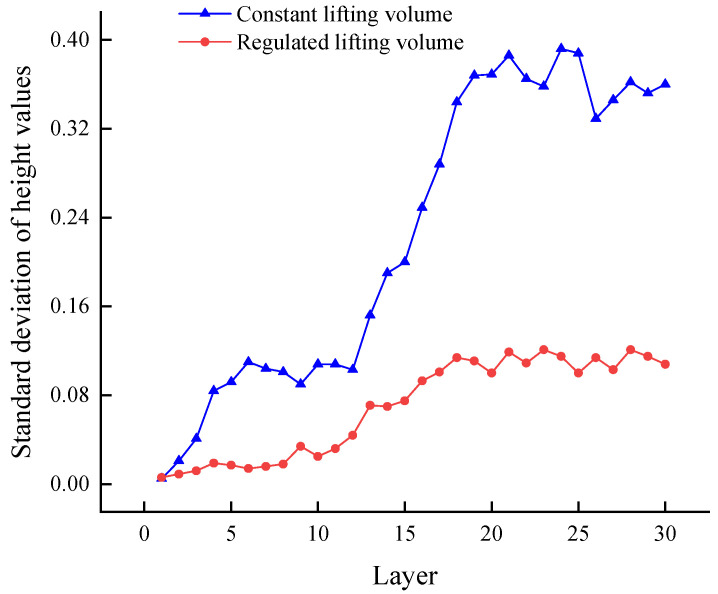
Height standard deviation for different experimental conditions.

**Figure 14 micromachines-14-01558-f014:**
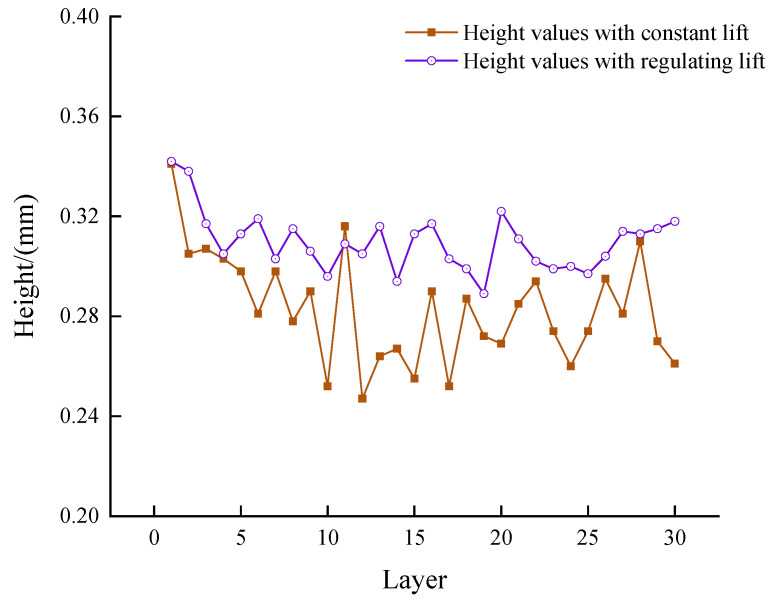
Height values for different experimental conditions.

**Table 1 micromachines-14-01558-t001:** 17-4PH chemical composition.

Element	C	Mn	Si	S	P	Cr	Ni	Cu	Nb
Wt%	0.07	1.0	1.0	0.025	0.035	15.0	3.0	3.0	0.15

**Table 2 micromachines-14-01558-t002:** Design of experiments.

Process Parameters (Symbol, Unit)	Value
Laser power (P, W)	1800
Scan speed (v, mm/s)	12
Argon gas flux (Q, L/min)	8
Laser spot diameter (d, mm)	6 × 2
Number of layers of cladding (n, /)	30
Powder feed rate (f, g/min)	15

## Data Availability

Not applicable.

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
