# Peer review of "Z-Increments Online Supervisory System Based on Machine Vision for Laser Solid Forming"

_micromachines, 2023, doi:10.3390/mi14081558_

Round 1
Reviewer 1 Report
Wang et al. present a comprehensive study on lasers machining, focusing on the theoretical and experimental aspects. They address the issue of improper Z-increment in laser solid forming, which can cause fluctuations in off-focus amounts during manufacturing, affecting the precision and quality of the fabricated component. To resolve this problem, they propose a closed-loop control system based on machine vision monitoring.
Their system utilizes edge detection technology and an inverse perspective transformation model to achieve real-time monitoring of the precise cladding height. By continuously assessing the cladding height, they can control the Z-increments in real-time, ensuring a constant off-focus amount throughout the manufacturing process.
Experimental results demonstrate that the proposed approach yields a maximum relative error of 1.664% in determining the cladding layer height, enabling accurate detection of this parameter. Additionally, real-time adjustments of the Z-increment quantity lead to reduced standard deviations of individual cladding layer heights and increased height of cladding layers. This proactive adjustment significantly enhances the manufacturing process's stability and improves powder material utilization.
Overall, the study provides effective guidance for process control and product optimization in laser solid forming, making it a valuable contribution and recommended for publication in Micromachines.
Reviewer 2 Report
This paper discussed on measuring the height using machine vision during 3D metal printing. The topic is practical and interesting, and is a fusion technique of the machine vision, 3D printing and precise control. The authors also experimented their method using industrial machines and devices. However this paper needs to be improved because of following reasons.
1. Chapter 2.1 is also the main contribution of this study as well as the edge detection. The authors used the IR range to acquire images, however the reason is not shown. It is also better to describe the reason using the spectral ranges of the laser, CCD camera and optical components.
2. Eqation (1) seems to be the rigid body transformation for camera tilt and offset from base point for calibration. However, Fig. 1 looks like the 3D detection of a target object, which causes confusing. You'd better to describe the reason why the rigid body transformation was applied.
3. Equation (2) involves the focal length, which indicates 3D surface of the target object variation on Z direction. However, defocusing problem on an acquired image will be caused, thus you should show how to solve the defocus in the image.
4. Fig. 1 and others should be centered and spaced. According to Fig. 2, Fig 1 should be rotated on the CCW direction. In those figures, cladding direction was not shown. According to Fig. 2, the camera is directed on a cross-section of stacking, thus perspective transformation seems to be minor.
5. Fig. 3. It is difficult to understand the relations among the coordinates and the focal direction. Thus, it is also hard to figure out why the equations are applied. A target object, camera and general concepts should be shown, because it is not clear what the authors intend to detect. The definitions of the variables must also be defined.
5. In Equation (2) and others, Zc is confusing. For example,
Zc[x y 1] and Zc[u v 1] are correct? Zc is the coordinate of P(Xc,Yc,Zc)?
Zc in equations (8) and (9) also should be checked.
6. You showed the values of M1, R and T, but they are less important. In camera calibration, it is better to show the calibration chart and algorithm. Further, I'd like to ask you that you used single of multiple points during calibration.
7. Furthermore, how did you calibrate the lens distortion? It is important for the projection algorithm and 3D relations because the authors assembled multiple lenses and optical components.
8. In chapter 2.4, Sobel, median and Otsu filters are well-known methods, so it is better to be described briefly in the text. The values shown in the text is unnecessary. Instead, it is better to show the image variance and silhouettes of rough metal edges according to those filters. The edge detection algorithm using those filters should be shown in a flow chart for easy reading. Definitions of Gx, Gy, G45 and G135 needs to be defined.
9. The test target has flat surface, but actual 3D targets has curved and complex. Do you have any solutions and discussions on the problems using the machine vision?
10. Check 'the' and 'a' in the text. For example, a off-axis -> an.
Round 2
Reviewer 2 Report
Thank you for your responses.
The authors properly answered most of the comments. One thing I'd like to point out is Figure 3.
The authors applied the perspective transform using a focal length in equations (2) and others. However, it is difficult to understand the coordinate relations and the focal length using Figure 3. It is better to show the focal length and the cladding layer in Figure 3.
